# When a Ciliate Meets a Flagellate: A Rare Case of *Colpoda* spp. and *Colpodella* spp. Isolated from the Urine of a Human Patient. Case Report and Brief Review of Literature

**DOI:** 10.3390/biology10060476

**Published:** 2021-05-27

**Authors:** Vlad S. Neculicioiu, Ioana A. Colosi, Dan A. Toc, Andrei Lesan, Carmen Costache

**Affiliations:** 1Department of Microbiology, Iuliu Hațieganu University of Medicine and Pharmacy, 400349 Cluj-Napoca, Romania; neculicioiu.vlad.sever@elearn.umfcluj.ro (V.S.N.); icolosi@umfcluj.ro (I.A.C.); anca.costache@umfcluj.ro (C.C.); 2Pneumology Department, Iuliu Hațieganu University of Medicine and Pharmacy, 400332 Cluj-Napoca, Romania; lesan.andrei@umfcluj.ro

**Keywords:** ciliate, flagellate, *Colpoda*, *Colpodella*, urine

## Abstract

**Simple Summary:**

In the era of new emerging diseases, particularly seen in patients with impaired immunity, it is of outmost importance to recognize unusual etiologic agents and to provide solutions regarding their treatment and prophylaxis. Our paper presents the first recorded evidence of the parasites *Colpoda* spp. and *Colpodella* spp. isolated together from the urine of a human patient. Although the patient did not experience any urinary symptoms and we discovered the parasites purely incidental, their disappearance was noticed after a combined therapy with two antibiotic drugs. In order to better understand the involvement of these two parasites in human pathology, we performed a brief review of the existing medical literature. Isolation of these parasites was recorded in different areas of the globe; however, we encountered a discrepancy in the diagnostic techniques used to identify these parasites. In medical parasitology molecular techniques provide the most accurate diagnostic but optical microscopy diagnosis, based on morphologic description of the parasites is also a useful, accessible and affordable diagnostic tool and it should not be neglected in cases involving rare parasites, where molecular diagnosis is not wildly available.

**Abstract:**

An often-overlooked side of the population aging process and the steady rise of non-communicable diseases reflects the emergence of novel infectious pathogens on the background of an altered host immune response. The aim of this article was to present the first record of a ciliate and flagellate protozoa recovered from the urine of an elderly patient and to review the existing medical literature involving these parasites. A 70-year-old female patient was admitted for breathing difficulties on the basis of an acute exacerbation of COPD (Chronic obstructive pulmonary disease) with respiratory insufficiency. The patient reported a long history of multiple comorbidities including COPD Gold II, chronic respiratory insufficiency, chronic heart failure NYHA III (New York Heart Association Functional Classification), type 2 diabetes and morbid obesity. During routine examinations, we ascertained the presence of two unusual protozoa, a ciliate and a flagellate, in the patient’s urine samples, identified on morphological criteria to be most likely *Colpoda* spp. and *Colpodella* spp., with similarities to *C. steinii* and *C. gonderi*. The presence of these parasites was not associated with any clinical signs of urinary disease. Following a combined treatment with ceftriaxone and metronidazole, we observed the disappearance of these pathogens upon discharge from the primary care clinic. This study highlights the importance of including unusual pathogens in the differential diagnosis of cases which involve immunosuppression.

## 1. Introduction

Human health is influenced by numerous factors, some under our control and some not. Global warming, aging of the population [1], increased travel between the continents, poverty, the steady rise of multiple chronic diseases such as obesity [2] and diabetes [3] in the context of the recent epidemiological shift [4] and the increasing usage of dexamethasone in the treatment of COVID-19 are all factors that determine the emergence of novel pathogens on the background of an altered host immune response.

Two examples of protozoa that have been viewed until recently as unable to infect humans are *Colpoda steinii* and *Colpodella gonderi*.

*Colpoda steinii* is a ciliate protozoan that was first discovered by Maupas in 1883 belonging to the family *Colpodidae*, order *Colpodida*, class *Colpodea*, phylum *Ciliophora* [5]. Ciliates from the class *Colpodea* are widespread around the globe and can be found in diverse habitats ranging from terrestrial/semiterrestrial (mosses, soil, bark of trees etc.) to aquatic environments (both fresh water—ponds, lakes, running waters and saltwater) [6]. *Colpoda steinii* has a simple lifecycle between two morphologic forms: mobile trophozoites and cysts. The cysts provide resistance to the external environment for prolonged periods of time and have been found even in the Siberian permafrost [7]. *Colpoda steinii* is considered to be non-pathogenic to humans but in spite of this fact several instances are cited in the literature that present cases of this parasite being present in urinary tract infections while no other microorganism was proven to be at the origin of the signs and symptoms exhibited by the patient [8,9,10].

*Colpodella gonderi* (Foissner and Foissner, 1984) is a flagellated protozoan belonging to the order *Colpodellida* (Cavalier-Smith, 1993) [11]. Protozoa from the genus *Colpodella* are free-living protozoa and the closest genetic relatives of Apicomplexans (phylum which includes important human pathogens such as *Plasmodium falciparum*, *Cryptosporidium parvum*, *Toxoplasma gondii*). *Colpodella gonderi* is considered a predatory flagellate and the literature cites multiple species of ciliates as being predated by this flagellate, including *Colpoda steinii* [12]. This protozoan is considered to be non-pathogenic to humans but two studies published in the last years cite different types of infection when no other proven pathogen was present except this parasite. The aforementioned studies describe an infection with similarities to *Babesia* spp. infections and a tick-borne case in which the patient presented neurological symptoms [13,14].

To the best of our knowledge, no scientific accounts are present in the literature citing the presence of both *Colpoda* spp. and *Colpodella* spp. in any biological human samples.

## 2. Case Presentation

We report the case of an elderly female patient, aged 70, who was admitted to the pneumology hospital in Cluj-Napoca, Romania, with the main complaint of breathing difficulties, dyspnea with orthopnea that started a few hours before admission. Family history was not significant but the patient has a personal history of multiple chronic diseases including COPD Gold II, chronic respiratory insufficiency with oxygen therapy at home, chronic heart failure NYHA III, type 2 diabetes with insulin treatment and class 3 severe obesity (BMI = 44.1, Body mass index). The initial examinations in the ER (emergency room) revealed respiratory acidosis with hypoxemia and hypercapnia (pH 7.349, pCO_2_ 56.2 mmHg, pO_2_ 65.5 mmHg and SO_2_ 82%), normal blood pressure and heart rate (Blood pressure = 120/80 mmHg, heart rate = 84 bpm). Multiple other examinations were performed including pulmonary X-ray that showed no significant radiologic abnormalities. The initial bloodwork showed only an abnormal level of blood glucose with values up to 450 mg/dL. The ER diagnostic was acute exacerbation of COPD with respiratory insufficiency and after stabilization the patient was transferred to the pneumology department.

The patient was hospitalized in the pneumology department for a total duration of 16 days, during which her condition steadily improved. Multiple routine tests were performed during her hospital stay, ranging from spirometry to bloodwork and urine tests.

Microscopic analysis of the urine sediment revealed the following results: 25–30 leukocytes/microscopic field, 8–10 squamous epithelial cells/field, rare bacterial flora and mobile parasitic trophozoites. The urine culture was negative.

Upon urine examination we managed to ascertain the presence of two distinct parasite genera, based on morphology and movement type. We identified trophozoites from the *Colpoda* genus, most probably *Colpoda steinii* according to the specific ciliate type mobility and following morphological characteristics: size ~40 × 20 μm (length and width), kidney shaped, covered with short cilia, cystotome cleft at about one-third of the distance from the anterior end, contractile vacuole at the posterior extremity, ovoid macronucleus and multiple food vacuoles. The morphology of one observed *Colpoda* spp. trophozoites is presented in Figure 1. The second identified parasite was a protozoan of the *Colpodella* genus, most probably *Colpodella gonderi*, with specific flagellate like movement patterns and the following morphological characteristics: size ~15 × 10 μm (length and width), egg shaped with an anterior pointed end, spherical nucleus localized relatively in the middle of the body, spherical inclusion at the end of the body, two flagella of an approximate length of 1.5–2 the size of the body. The morphology of multiple *Colpodella* spp. trophozoites is presented in Figure 2 and the movement patterns are presented in Appendix A. We measured the intensity of the infection by counting the number of observed trophozoites in multiple microscopic fields, using 40× objective. In the case of *Colpodella* spp. we observed ~20 trophozoites/field, as seen in Appendix A. *Colpoda* spp. trophozoites were observed in much smaller numbers.

Noteworthy is the fact that our patient did not have any other urinary pathology, nor did she exhibit any urinary symptoms before or during the hospital stay.

We managed to obtain multiple urine samples during the hospitalization of our patient and observed the following chronological evolution of urinary parasitic contamination. In the first sample we managed to ascertain the existence of 2 distinct parasite trophozoites, namely *Colpoda* spp. and *Colpodella* spp., based on the specific morphology and movement patterns. In the next samples collected in the following days, only one protozoal species could be found in the urine—*Colpodella* spp. Upon discharge from the hospital, we managed to obtain one final urine sample in which no parasite trophozoites could be found. We did not observe parasitic cysts in any of the analyzed samples.

The treatment of our patient focused mainly on the management of her chronic diseases and included both a pharmacological approach and non-invasive oxygen-therapy. Our patient was treated with 2 antibiotics during her stay, ceftriaxone and metronidazole, with different treatment durations. Ceftriaxone was administered for the duration of hospitalization, in the dosage 1 g/24 h, for the prophylaxis of potential bacterial infections. In addition, metronidazole was administered for 5 days, 250 mg every 12 h. The treatment with metronidazole was started as soon as the first urine analysis was interpreted and the parasites observed. The subsequent urine sediment examinations showed the disappearance of *Colpoda* spp. firstly, and upon discharge the disappearance of *Colpodella* spp.

## 3. Discussion

Several limitations were encountered in regards to the diagnostic workflow. Firstly, we were unable to perform a PCR assay for the identification of the observed protozoa. Lacking a nucleic acid identification assay, we had to rely solely on the morphological identification on wet mounts and Giemsa-stained smears. Secondly, we did not screen our patient for the intestinal or genital presence of this protozoa in order to exclude a contamination from this body sites. Thirdly, we did not screen other patients proactively for the presence of these parasites, but in the normal workflow of biological samples no other cases having these parasites were identified before, during or after the timeframe of hospitalization of our patient.

Before admitting that our patient had a rare urinary parasitic infection, we tried to rule out all other possibilities, the main one being the accidental contamination of the sample with the parasites. During this timeframe, urine sediment samples from other patients hospitalized in our clinic were analyzed according to the same protocol and no other sample tested positive for the presence of parasites. Urine samples were collected according to aseptic standards, in single use sterile plastic containers. We did not manage to screen our patient for the intestinal or genital presence of these parasites. The presence of parasite trophozoites in our patient’s urine was proven on multiple days, in different urine samples, with a decreasing number as expected in a patient treated with metronidazole. The last urine sample that was analyzed before discharge from the hospital tested negative for parasite trophozoites, presence of leukocytes or bacteria. During the course of this case, no concomitant bacterial urinary infection was observed in our patient, in addition to the protozoa.

Compared to the work of Costache et al. [10], we were unable to witness the process of encystation or excystation, nor did we see any formed *Colpoda* spp. cysts. The most plausible explanation for this fact could be related to the mistiming of the examinations with the life cycle of the parasite. Even though unlikely, two other explanations exist as well. Firstly, the combined treatment with ceftriaxone and metronidazole might have a better effect on *Colpoda* spp. compared to *Colpodella* spp., and thus, explaining the fact that we only managed to observe the ciliate in the first examination and not in the following days after starting the treatment. Secondly, although highly unlikely, a predation of *Colpoda* spp. by *Colpodella* spp. might have occurred, thus rendering the observation of ciliates impossible on further examinations. An argument for this hypothesis comes from the discrepancy in the numbers of the two parasites: we observed only a few *Colpoda* spp. trophozoites as compared to increased numbers of flagellates.

The novelty of our case comes from the observation of an associated urinary infection with two different protozoa, a ciliate and a flagellate and from the fact that there are no other cited cases in the literature in which *Colpodella* spp. was found in the urinary tract [13,14].

The source of contamination in the case of our patient can only be speculated upon. These parasites are often found in the environment and can easily contaminate any water body though dust. It is not inconceivable that our patient was contaminated by using rain water instead of tap water for bathing or simply by bathing with water contaminated by dust though an open window. While we managed to get thorough information about our patient’s medical history, relevant details about her living conditions and habits were left out.

The main differential diagnosis that we performed was to exclude the following pathogens: *Trichomonas vaginalis* (one of the most frequently recovered flagellates from urine samples) [15] and *Balantidium coli* (considered to be the only ciliate pathogenic to humans). No other parasites were found in the urine. Based on morphological characteristics and examination of the movement pattern, we excluded all the aforementioned pathogens.

Regarding the treatment, the evidence seems to suggest the concomitant treatment regimen with ceftriaxone and metronidazole to be effective against urinary contamination with *Colpoda* spp. and *Colpodella* spp.

We are aware of the coincidence of presenting a similarly rare case of urinary infection as previous work from Costache et al. [10] in Cluj-Napoca, Romania. This occurrence might suggest a common local link that requires further investigation in upcoming studies.

Given the rarity of cited cases in the literature, the presence of the second case in the same geographic location may sound weird. However, a possible explanation of this situation might be the increased reliance on microscopical examinations for urine and stool samples in Romania, as opposed to other European countries. Wide spread access to automated diagnostic systems may come with the downside of missing rare microorganisms present in different samples. Furthermore, the lack of accessible molecular diagnosis in some countries can only add to the potential confusion. Therefore, an increased probability of misdiagnosis or lack of identification of rare pathogens can be seen across the literature.

Further studies are required to assess the frequency of this kind of infections, to assess the risk factors, to determine the route of infection, to clarify the physiopathology and to propose standardized treatment regimens for them.

## 4. Review of Literature

We performed a review of the medical literature regarding the isolation of *Colpoda* spp. and *Colpodella* spp. from humans, searching on PubMed and Cochrane library electronic database, up to 15 March 2021. The results are summarized in Table 1. We considered the following terms included in the studies title or abstract: “colpoda”, “colpodella” combined with the Boolean operator” AND” along with” human”,” urine”,” blood” and ”diagnostic”. We excluded studies written in languages other than English and French. We found a selection of five studies, four case reports and one original article that we included in our review.

Regarding *Colpodella* spp., we found two case reports both from China and both patients were females in their fifties. Yuan C et al. reported a case of *Colpodella* spp. recovered from the blood of a patient with hemolytic anemia and immunosuppression due to solitary natural killer cell deficiency who responded well to the treatment with atovaquone and azithromycin for 8 weeks [13]. Jiang J et al. reported a case where the parasite was found in the cerebrospinal fluid of a patient with neurologic symptoms who was treated with doxycycline [14].

Since there are just two case reports of this parasite isolated from humans it is difficult to draw a conclusion in terms of pathological involvement and treatment. However, regarding the diagnostic, in both cases researchers used 18S rRNA PCR for molecular diagnosis. Unfortunately, molecular diagnostic techniques are still not available at large scale so we want to emphasize the importance of morphologic diagnosis of *Colpodella* spp. The most important morphologic features are presented in Table 2.

Regarding *Colpoda* spp., we found 3 articles, two case reports and one research article. The case reported by Costache C et al. from Romania [10] and the case reported by Guy Y et al. from Algeria [8] are describing the parasite *Colpoda* spp. isolated from urine and identified based on morphologic characteristics. On the other hand, Bouchoucha I et al. [16] performed an analysis of several contact lens solutions, recovered from patients with the clinical diagnosis of keratitis and corneal ulcer and discovered the presence of the parasite *Colpoda steinii* in one specimen of contact lens solution. In that study, the diagnostic of the ciliated protozoan was performed using 18S rRNA PCR. However, this study shows a second source of potential interaction between humans and the rare ciliate, *Colpoda* spp.

Even though *Colpoda* spp. is not considered a human pathogen, in all the case reports from the literature that we included in the review, this ciliate parasite was isolated from the urine of the patients. Further research is required to establish if this parasite is able to survive in the urinary bladder of healthy individuals or just in that of immunosuppressed ones.

There is an established link in the literature concerning immunosuppression and the increased incidence of bacterial or protozoan infections. Including the case presented by us, in four out of the six case reports included in our review the patients presented a form of immunosuppression. The presented evidence points to the existence of a plausible link between immunosuppression and infections caused by unusual pathogens.

In the case of rare parasites, even if molecular diagnosis represents the most accurate tool, optical microscopy diagnosis based on morphologic criteria should not be neglected.

## Figures and Tables

**Figure 1 biology-10-00476-f001:**
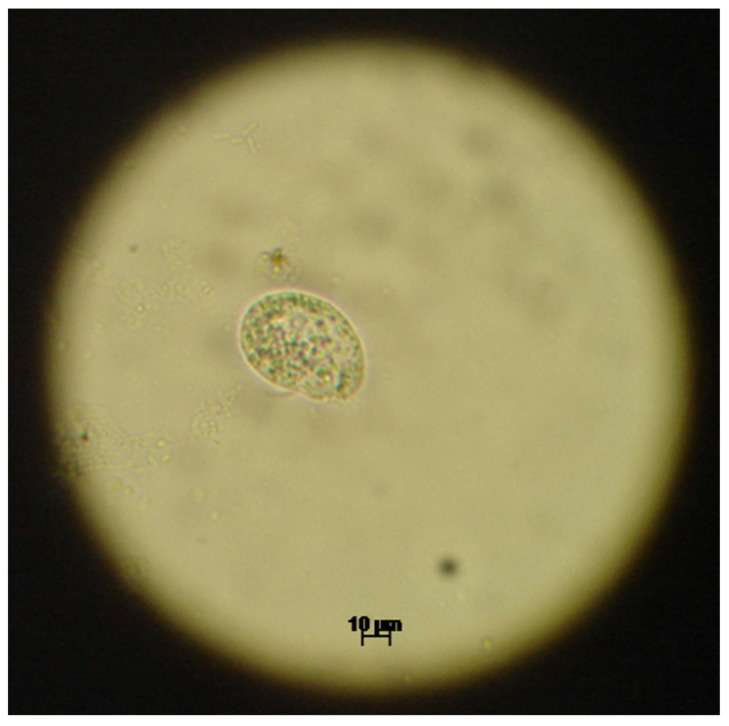
The optical microscopy image (40×) of the *Colpoda* spp. trophozoite.

**Figure 2 biology-10-00476-f002:**
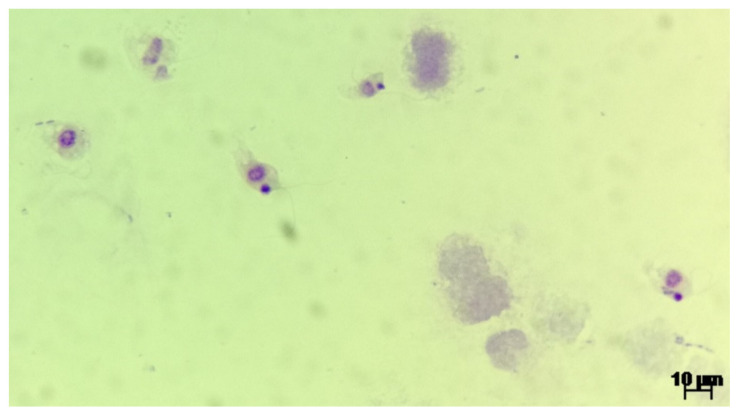
The morphology of *Colpodella* spp. trophozoites in a Giemsa-stained smear (100×).

**Table 1 biology-10-00476-t001:** A selection of studies involving the diagnosis of *Colpoda* spp. and *Colpodella* spp.

Reference	Year	Country	Study Design	Pts (No.)	Age	Gender	Parasite	Sample	Underlying Condition/Predisposing Factors	ID	Hospitalization	Treatment	Outcome
This study	2021	Romania	Case Report	1	70	F	*Colpoda* spp. and *Colpodella* spp.	Urine	COPD, Obesity, Diabetes	OM	/	ceftriaxone, metronidazole	FR
Jiang J et al. [14]	2018	China	Case Report	1	55	F	*Colpodella* spp.	Cerebrospinal fluid	N/A	MD	10 days	doxycycline	/
Bouchoucha I et al. [16]	2016	France	Research Article	1	/	/	*Colpoda steini*	Contact lens solution	N/A	MD	/	N/A	/
Yuan C et al. [13]	2012	China	Case Report	1	57	F	*Colpodella tetrahymenae*	Blood	Hemolytic anemia. Solitary natural killer cell deficiency	MD	8 weeks	atovaquone, azithromycin	FR
Costache C et al. [10]	2011	Romania	Case Report	1	36	M	*Colpoda* spp.	Urine	Psychiatric disorder, Homeless	OM	2 weeks	doxycycline, metronidazole	FR
Guy Y et al. [8]	1968	Algeria	Case Report	1	57	M	*Colpoda* spp.	Urine	Atrial Fibrillation, Gout	OM	/	/	/

Pts: patients; FR: full recovery; M: male; F: female; OM: optical microscopy; MD: molecular diagnosis; ID: identification; /: not reported.

**Table 2 biology-10-00476-t002:** Morphologic characteristics of *Colpoda steinii* and *Colpodella gonderi* trophozoites.

	Trophozoites	
	Colpoda Steinii	Colpodella Gonderi
~ Size μm (length/width)	40/20	15/10
Shape	Kidney shaped	Egg shaped
Motility structures	Cilia (body covered with short cilia)	Flagella (two flagella of 1.5–2 the size of the body)
Nucleus	Ovoid, macronucleus	Spherical, centered
Other characteristics	Cystostomal cleft at about one-third of the distance from the anterior end contractile vacuole at the posterior extremity multiple food vacuoles	An anterior pointed end spherical inclusion at the end of the body

## Data Availability

Not applicable.

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
