# Peer review of "When a Ciliate Meets a Flagellate: A Rare Case of Colpoda spp. and Colpodella spp. Isolated from the Urine of a Human Patient. Case Report and Brief Review of Literature"

_biology, 2021, doi:10.3390/biology10060476_

Round 1

Reviewer 1 Report

The case report titled “When a ciliate meets a flagellate: A rare case of Colpoda spp. 2 and Colpodella spp. isolated from the urine of a human patient. 3 Case report and brief review of literature” is well-written. It is indeed the first case report about both ciliate and flagellate presented in human urine. Thus, grant its publication in “biology” after revision.

  1. It is not clear to me if the urinary protozoa infection is related to the patient’s symptoms. Please consider including a timeline of the clinical course to also facilitate the discussion in Line165-176.
  2. Section2, case presentation: please provide the intensity of the protozoa infection if measured.
  3. Figure2, please state the difference of the 3 pictures included. Otherwise, it is good to have only 1 picture.
  4. Supplement Video, please provide more details of the video including the magnifying power.
  5. Table1.
    • Besides the tick-born case (Jiang J et al., 2018), is it true that patients in other reports are immune-deficient? It will be helpful to include the patients’ immune status.
    • Please clarify “Duration”. Is it the observation duration, symptom duration, or treatment duration?
    • The review of the literature would be more informative if the authors include this report: Katsuhiko M, Kaoru K, Masako I, et al. Two cases of the Ciliate protozoa Colpoda spp. detected from urine. Japanese Journal of Medical Technology. 2006;55(5):644–648
  6. Line225, typo of “to” : …… cu draw a conclusion.
  7. Please consider not to use “our study” in the manuscript since it is the presentation and discussion of a case, not a designed research.  

Author Response

We want to thank you for the input and evaluation of our manuscript.

Please see attachment for a point-by-point response to the suggestions made.

Kind regards,

The authors

Reviewer 2 Report

In this case report, the authors report the unusual finding of a ciliate and a flagellate organism in the urine of a patient with several comorbidities, including COPD.

The main limitation of the article is that no molecular identification of the protozoan organisms was performed. I would therefore refrain from referring to them as C. steinii and C. gonderi on the basis of morphology only. They should be referred to as Colpoda sp. and Colpodella sp., with morphologic similarity to C. steinii and C. gonderi.

Furthermore, it should be clearly stated (already in the Abstract) that the infection was not associated with any clinical signs of urinary disease. The medical relevance is thus questionable. This leads to my second suggestion, namely to significantly shorten this report.

The literature review presented at the end of the manuscript can easily be incorporated into the Introduction/Discussion, and does not need a separate section. Furthermore, the first three paragraphs of the Introduction dealing with the increased prevalence of non-communicable diseases and immunosuppressive disorders should be considerably shortened in relation to the significance of the study’s findings.

I missed some speculation regarding possible sources of the infection/transmission modes.

Minor comments:

Please write genus and species names in italics throughout the manuscript.

l. 49: I assume you mean infectious diseases instead of non-infectious diseases in this sentence.

L. 151: This sentence is confusing. While accidental contamination should of course be ruled out, there is certainly no danger of mistaking these protozoa for bacteria. It should rather be stated that no concomitant bacterial infection was present in addition to the protozoa.

L. 160: It is a pitty that no stool sample was obtained for a coproscopic analysis. Was the presence of the organisms in the genital tract excluded?

LL. 184-191: This paragraph can be reduced to simply stating that no other parasites or parasitic stages were found in the urine. It is not necessary to state that Schistosoma haematobium or Enterobius sp. were excluded, because these parasites or their eggs would look completely different and cannot be confused with these protozoa.

Table 1: It is very unusual and confusing to place the line headings at the right side of the table, please move them to the left.

Figure 2 is of low quality. At least the lower two panels should be replaced by better-resolution photographs.

Author Response

(The authors gave the same response as above.)

Round 2

Reviewer 2 Report

The manuscript has improved compared to the previous version. I still think that the “Review of the literature” does not require a separate section – after all, there are only three previous reports of Colpoda spp. and two reports of Colpodella sp. in human biological samples. However, I would leave the decision on this matter to the responsible editor.

L. 131: It should be stated here that intestinal and vaginal presence was not excluded.

L. 137 (previously L. 151): The authors stated in their response letter that they modified this section, however, the text is still the same in the revised manuscript. (My previous comment: L. 151: This sentence is confusing. While accidental contamination should of course be ruled out, there is certainly no danger of mistaking these protozoa for bacteria. It should rather be stated that no concomitant bacterial infection was present in addition to the protozoa.)

The structure of the discussion can be improved, I would move the paragraph on differential diagnoses after the paragraph dealing with possible contamination (i.e. after L. 151), and place the paragraph on possible sources of infection as well as on the treatment at the end of the discussion.

L. 218/219: This sentence is not required here, because it is described in detail in the previous sections. If keeping this section on review of the literature, please remove details of the present case report to avoid repetition and concentrate only on previously published reports.

L. 236: It is not necessary to cite the study regarding bacterial infections here, as there is ample evidence regarding a link between immunosuppression and protozoan infections – think of Toxoplasma or Cryptosporidium, for example.

Table 1: In some of these studies, molecular identification of the parasites was performed. So, if identification down to species level was possible in any of these studies, please state the species diagnosed and not only the genus.

Author Response

We want to thank you for all the hard work.

Please see the attachment for a point-by-point response to the comments.

Kind regards

The authors.
